# Endurance Exercise Mitigates Immunometabolic Adipose Tissue Disturbances in Cancer and Obesity

**DOI:** 10.3390/ijms21249745

**Published:** 2020-12-21

**Authors:** José Cesar Rosa-Neto, Loreana Sanches Silveira

**Affiliations:** 1Immunometabolism Research Group, Department of Systems Biology, Institute of Biomedical Sciences 1, University of São Paulo (ICB1-USP), São Paulo 05508-000, Brazil; loreana_loly@hotmail.com; 2Laboratory of Experimental Surgery, Department of Surgery, Clinics Hospital of the Faculty of Medicine, University of São Paulo (HC-FMUSP), São Paulo 01246-903, Brazil

**Keywords:** physical activity, immunometabolism, immune cell, low grade inflammation, adipocytokines

## Abstract

Adipose tissue is considered an endocrine organ whose complex biology can be explained by the diversity of cell types that compose this tissue. The immune cells found in the stromal portion of adipose tissue play an important role on the modulation of inflammation by adipocytokines secretion. The interactions between metabolic active tissues and immune cells, called immunometabolism, is an important field for discovering new pathways and approaches to treat immunometabolic diseases, such as obesity and cancer. Moreover, physical exercise is widely known as a tool for prevention and adjuvant treatment on metabolic diseases. More specifically, aerobic exercise training is able to increase the energy expenditure, reduce the nutrition overload and modify the profile of adipocytokines and myokines with paracrine and endocrine effects. Therefore, our aim in this review was to cover the effects of aerobic exercise training on the immunometabolism of adipose tissue in obesity and cancer, focusing on the exercise-related modification on adipose tissue or immune cells isolated as well as their interaction.

## 1. Introduction

Adipose tissue (AT) is a very complex organ which is able to expand or hypertrophy in overload of nutrients (stored as triacylglycerol) in adipocytes, or retract in periods of energy restriction by releasing fatty acids and glycerol in order to supply other body cells with energy. The AT is named according to its location in the body. The subcutaneous adipose tissue (SAT), found under the skin, and the visceral adipose tissue (VAT), that is distributed within the trunk along with the organs [1].

AT location is closely related to its function and consequently to its biology, thus a considerable difference between the different AT depots are found in animals and humans [2]. While the increase on the VAT depot is related with low chronic inflammation and insulin resistance, the SAT is associated with improvement on glucose homeostasis and it is not linked with chronic inflammation. In fact, the transplantation of SAT to VAT depots, in mice, improved the glucose homeostasis and decreased body fat mass [3]. On the same hand, AT distribution in human body is modulated by sexual dysmorphisms and aging, whereas the AT distribution in newborns is very different from elderly [3].

Noteworthy, many physiological functions are also fat depot-dependent. In 1991 Pond and Mattacks showed that both the ratio of stimulated-lipolysis by noradrenalin and the ability of insulin in blocking lipolysis are dependent on fat location [4]. The complex biology of AT can be explained by the diversity of cell types that compose this tissue. Adipocytes make up approximately 80–90% of AT, moreover adipocytes can be brown or white, while the stromal portion is formed by fibroblasts, pre-adipocytes and especially by resident and non-resident immune cells [5].

The white adipocyte is characterized by a big unilocular lipid droplet, few mitochondria and a higher ability to expand for triacylglycerol storage. On the other hand, the brown adipocyte has many small multilocular lipid droplets, great number of mitochondria and its main function is heat production by oxidative phosphorylation uncoupling [1]. In 2012 Wu et al. showed that in humans and rodents, white adipocytes that underwent a specific stimulation were able to modify their transcriptional pathways and function to a more like brown adipocytes, although they still had white adipocyte morphology [6]. Moreover, brown adipocytes can show similarities with white adipocytes such as lipid droplet enlargement and reduction on mitochondria numbers and thermogenic effects. This process is called “whitening” and is found in brown AT of mice fed with a high fat diet (HFD). An overexpression of induced carbohydrate element response-binding protein is also observed in this phenotype and can explain the role of higher insulin and glucose levels in obese animals as responsible for this change on brown adipocyte biology [7].

Most recently the importance of a third AT type—the so-called pink AT—was observed, especially in breast cancer. The adipocytes from pink AT are milk-secreting alveolar cells that suffer differentiation in WAT. Their morphology has robust cytoplasmatic lipid droplets, the apical region presents microvilli and the nucleus is located centrally within milk-containing granules [8]. The pink colour is observed during their transdifferentiation process in pregnancy and lactation [9].

With regard to the immune cells infiltrated in adipose tissue, it was observed that their type and proportion change under different conditions such as cancer and obesity [10]. In healthy adipose tissue the immune cell population is composed by a majority of cells with an anti-inflammatory profile, especially eosinophils, innate lymphoid cells (ILC2), CD4+ Th2 lymphocytes subset, T regulatory lymphocyte and M2 macrophage. On the other hand, this proportion is deeply modified in sterile inflammation diseases (obesity, cancer, diabetes and cardiovascular diseases) [11].

In obesity, the immune cell proportion is characterized by an increase in the number of M1 macrophages, CD4+ Th1 lymphocytes and CD8+ cytotoxic T lymphocytes (CTLs) [12]. In AT from HFD mice immune cell recruitment starts three days after the diet begins. The kinetics of immune cell infiltration are not fully clear. Talukdar and colleagues showed that the first immune cells recruited are neutrophils, which in turn, increase the chemoattraction of macrophages [13], Nishimura and colleagues showed that the first immune cells infiltrated in inflammation are CD8 T lymphocytes with a reduction of T regulatory CD4 lymphocytes and a posteriori recruitment of macrophages [14]. Indeed, with this great interaction between immune cells and adipocytes, the AT can be considered not only a metabolic but an immune organ as well [15]. Therefore, the balance between Th1, T regulatory and Th2 lymphocytes [16] and M1 and M2 macrophages [17] is essential to homeostasis of AT. An elegant study demonstrated that metabolic disturbances induced by elevated body mass index are associated with insulin resistance, inflammation and dyslipidemia. Whereas the M1:M2 ratio is not a predictor, howsoever the role of a subset of macrophages may be important to remodeling of AT [18]. In 2011, Mathis and Shoelson introduced the new emerging branch of biological science called immunometabolism [19]. Immunometabolism is a two way road, with interactions between metabolic tissues, such as adipose tissue, liver and skeletal muscle and immune cells, along with the importance of metabolic pathways in immune cell differentiation [20]. Therefore, the understanding of AT biology on health and diseases is essential to discovery new mechanisms and approaches to treat immunometabolic diseases, such as obesity, diabetes, cancer and cardiovascular disorders.

Physical exercise is an excellent tool for the prevention and adjuvant treatment of metabolic diseases [21]. On the other hand, a sedentary lifestyle is considered a disease, as the lack of physical activity induces visceral AT hypertrophy and triggers systemic inflammation which increases the risk of chronic diseases [22]. Physical exercise is able to modify metabolic pathways during and after sessions by inducing considerable immunometabolic changes [23,24].

Aerobic exercise training (AET)results in pleiotropic effects on adipose tissue and immune cells, with many variables as cited above and the results may vary among species (man, rat and mice) and type of adipose tissue (brown or white). Our aim in this review is therefore to show the effects of aerobic exercise training on the immunometabolism of adipose tissue in obesity and cancer.

## 2. The Role of Aerobic Exercise Training in Adipose Tissue Immunometabolism

White adipose tissue (WAT) shows many physiological functions, including the storage of lipids for fatty acid supply in a state of energy deprivation and/or during aerobic exercise. Aerobic exercise training can increase mitochondrial activity resulting in browning. Moreover, fatty acid and lipid composition are modified and there are also alterations on the profile of immune cells infiltrated in WAT and changes in the pattern of adipokines produced and released into circulation [25].

In mice, adaptations and modifications on subcutaneous WAT such as increase in mitochondrial gene expression and raise of oxygen consumption rate were observed on the eleventh day of wheel cage running [26,27]. It is not fully clear if AET is able to induce the browning on adipose tissue. In rodents the long-term AET (more than 8 weeks) consists of SAT browning [25]. Furthermore, three weeks of wheel voluntary running induces browning on inguinal SAT [28]. However, in humans, browning of SAT was not observed. In healthy humans 10 weeks of AET did not show SAT browning, although the trained subjects showed increased expression of insulin receptor, hexokinase II and succinate dehydrogenase, suggesting that the insulin sensitivity and oxidative metabolism are raised after this endurance training duration [29]. Moreover, well trained athletes from AET did not show an increase in SAT browning in comparison with sedentary subjects [30].

The AET adaptations on BAT are still controversial. Less current studies showed that AET (swimming or running) in rodents increased the mitochondrial activity in BAT [31,32]. However, more recent research showed BAT mass is reduced after 8 weeks of AET [33]. In the same direction, well trained athletes had decreased glucose uptake and cold-induced BAT activation compared with sedentary subjects [30].

The AET induces adjustments on adipocytokines released and the most studied in this context are adiponectin and leptin. AET reduces the mRNA and protein expression of both adipocytokines in SAT [34]. On the other hand, other studies showed that AET increased cytokines in animals. Lira et al. showed that IL-10 and TNF-α levels were raised in mesenteric adipose tissue (MEAT), and no difference was found in the retroperitoneal depot. Although both cytokines increased in MEAT, the elevation on IL-10 was higher than TNF-α, resulting in high IL-10/TNF-α ratio [35]. In the same direction, the epididimal and subcutaneous fat depots showed elevation of cytokines mRNA [34,36]. Recently, we observed that 8 weeks of moderate (40–60min, 5 times week at 55–65% of maximal velocity) treadmill running reduced TNF-α protein on subcutaneous WAT [37]. Finally, the training load is important for the inflammatory response in WAT. It is well established that moderate training induces a more pronounced anti-inflammatory profile. In this sense Lira et al. showed that overtrained rats presented increased TLR-4-NFkB pathways in WAT [38].

The role of AET over the immune cells infiltrated in WAT is still little studied. In general, it is observed that WAT of lean subjects show high concentrations of eosinophils, Treg lymphocytes, and Breg lymphocytes in humans and mice, furthermore lean adipose tissue from mice presents M2 macrophages [39]. Recently, it was demonstrated that lean trained mice possess a higher proportion of M2 macrophages (CD11c^+^; CD206^+^) than sedentary lean mice and that it is independent on peroxisome proliferator-activated receptor gamma (PPAR-γ) expression in myeloid cells [37]. PPARγ is a transcriptional factor that regulates immunometabolic pathways in adipose tissue. It is highly expressed on adipocytes and controls adipogenesis, de novo lipogenesis and insulin sensitivity. PPARγ is also expressed in immune cells, in special myeloid cells and its activation regulates the M2 polarization [40]. Nonetheless, as demonstrated the exercised-induced anti-inflammatory profile in SAT like the reduction in M2 polarization is PPARγ-dependent in myeloid cells [41].

So far, we have shown the effects of AET on WAT, therefore, in the next sections, we will focus on the role of AET in WAT in diseases like cancer and obesity that are characterized by low-grade inflammation, in which WAT participates in the genesis or maintenance.

## 3. Obese Adipose Tissue Remodeling by Aerobic Exercise Training

WAT is in constant remodeling, and depending on the excess or deprivation of energy it shows a high but not infinite ability to expand. Furthermore, this expansion can be by hypertrophy or hyperplasia. Hypertrophic process is a raise on the triacylglycerol storage into the lipid droplets with an increase on cell size, while hyperplasia is an increase in the number of adipocytes that are differentiated from pre-adipocytes [42].

In addition, the adipocyte molecular machinery is able to respond to acute modifications ofthe supply of nutrients. In this sense, adipocytes have a great number of hormone receptors, especially insulin and adrenergic receptors with robust sympathetic innervations [43], so the balance between insulin and adrenergic response regulates the storage or released of fatty acids. Recently, Fitzgibbons discussed the role of insulin and catecholamine in WAT. Both hormones increase the blood flow in WAT for different needs, while cathecolamine-induced vasodilatation triggers free fatty acids release, the insulin-induced increases blood flow to deliver the postprandial excess of glucose to the insulin-sensitive organs [44]. Thus, insulin resistance on adipocytes impairs the O_2_ delivery and can create a low pO_2_ environment. It is interesting that SAT vascularization is higher compared to visceral depots and it is correlated with better insulin sensitivity in humans [45].

Obese people have drastic modifications on the patterns of adipokines and cytokines production by WAT [43]. In this sense many cytokines and adipokines have interplaying functions between immunological/inflammatory and metabolic response. The best example is TNF-α, a cytokine usually found in high levels in WAT and blood of obese subjects. This cytokine induces to a pro-inflammatory response through disruption of adipogenic signals and insulin sensitivity due to PPARγ blocking and increased lipolysis [46]. Thus, an obesity-related immunometabolic disturbance is caused by nutrient overload (lipotoxicity and glucotoxicity) and by a deep perturbation of adipocytokines production by different WAT depots, which results in local and systemic implications [47].

In this sense, AET is the most suitable tool to mitigate these immunometabolic disturbances and reduce the risk to co-morbidities associated with obesity. AET increases the energy expenditure, reduces the nutrition overload and modifies the profile of adipocytokines and myokines with paracrine and systemic effects [48,49]. Moreover, AET is able to induce the remodeling of lipid species in SAT and BAT. Phospholipids and triacylglycerol were altered after three weeks of voluntary running. This interesting find showed that AET promotes a modification on the pattern of lipids and it is dependent on adipose tissue depot [50]. However, the physiological effects of these changes on lipid species should be better investigated to understand the function of each lipid class in the adipose tissue remodeling by AET. The effect of exercise over the immunometabolism depends on type, mode, duration and the practitioner’s fitness level.

## 4. Aerobic Exercise Training Causes Immunometabolic Adaptations That Mitigate the Disturbances Caused by Obesity

### 4.1. Findings in Rodent Models

Rodent models are necessary to understand the molecular mechanisms by which AET improves the immunometabolic disturbances caused by obesity. Studies with animal models easily isolate exercise training effects without caloric restriction, whereas in most human studies it is impossible to distinguish the isolated effects of exercise, diet and weight loss in metabolic and immunological improvement.

In obesity, the huge increase in fatty acids delivered together with the limited ability of adipocyte hyperplasia and consequently in restricted blood flow results in signaling of danger-associated molecular patterns (DAMPs) in WAT, concomitant with metabolic and inflammatory stress [23]. The mitochondrial stress caused by fatty acid oversupply releases many DAMPs such as mitochondrial DNA, ATP and cytochrome C [51]. The DAMPs’ activate pattern recognition receptor (PRR) pathways that incite the inflammasome machinery. Thus, the NLPR3 inflammasome pathway is activated and caspase-1 cleaves pro-IL-1β and pro-IL-18 in IL-1β and IL-18, respectively [52]. These pro-inflammatory cytokines together with monocyte chemoattractant protein (MCP-1) released into adipose tissue trigger the recruitment of many immune cells with stimulus to differentiation into pro-inflammatory subsets, thus, triggering low-grade inflammation [23].

Mice fed for four weeks with HFD were trained for 10 weeks on a treadmill at moderate intensity, five times per week. These mice were maintained on HFD and the AET was able to reduce the TNF-α and IL-18 protein expression on WAT, together with a reduction in weight gain [53]. Yamashita et al. showed that the anti-inflammatory effects of AET in obese rats were WAT depot-dependent. While caloric restriction (80% of daily caloric intake) reduced the cytokines content in retroperitoneal and MEAT, AET reduced the pro-inflammatory cytokines on MEAT only [54]. However, the improvement on glucose homeostasis was similar between caloric restriction and AET groups, highlighting the association between inflammation of MEAT and metabolic disturbance in obese rats. Similarly, mice fed with HFD (mainly *trans*-fatty acids) for 12 weeks and submitted to 8 weeks of AET showed a reduction on TNF-α and MCP-1 in SAT. Moreover, IL-6 and adiponectin protein expression were attenuated in isolated adipocytes but not in AT (stromal plus adipocytes). Thus, the contribution of adipocytokines in obesity and the AET effects seems to be different between WAT depots and isolated adipocytes [41]. In a study that fractionated adipocyte size and evaluated cytokines secretion, it was concluded that the very large adipocytes are the responsible for promoting inflammation [55], suggesting that VAT and SAT may play different roles in adipocytokines secretion and consequently in local inflammation. Additionally, chronic diseases and diagnosed pathologies are able to further reduce adiponectin levels and exercise does not seem to be an effective tool for restoring adiponectin levels and body composition changes mediated by exercise programs may be the most important outcome for adiponectin modulation [56].

The AET effects are not only in production of adipocytokines from adipose tissue, but it also induces pleiotropic changes in different systems and organs of the body. Many benefits found in adipose tissue are a consequence of the production of circulating factors released by skeletal muscle, liver, brain and other organs. Fibroblast growth factor 21 (FGF-21) is a strong candidate to be responsible for mediating the glucose whole body homeostasis by effects of glucose uptake in adipose tissue [57]. AET was able to reverse the FGF-21 resistance caused by obesity, specifically in AT, with subsequent improvement of glucose metabolism. Moreover, the knockout model for FGFR-1-KLB pathway on adipocytes showed impaired AET benefits [58,59].

As aforementioned the subset of immune cells infiltrated on WAT impacts tissue homeostasis. Macrophages are the majority of immune cells in the stromal portion of WAT (approximately 60%) [60]. Until the last decade it was believed that the main source of adipose tissue macrophages (ATM) was infiltration from circulating monocytes. However, macrophages from VAT are derived from primitive yolk-sack progenitors [61,62] together with the fact that there is a recruitment of monocytes to WAT in obesity [63].

AET is an excellent instrument to reduce obesity-induced macrophages infiltration in WAT. In 2010, Kawanish and colleagues showed that mice fed with HFD for 16 weeks showed increased macrophage infiltration and mRNA expression of M1 phenotype [64]. Recently, our group confirmed that 8 weeks of AET in obese mice caused a great reduction on M1 phenotype on SAT as analyzed by flow cytometry. Moreover, we showed that this reduction on M1 marker is PPAR-γ-dependent on myeloid cells (PPAR-γ Lys-CRE), since mice with this specific deletion sustained the proportion of M1 subsets on subcutaneous WAT even when trained [41].

An important point to be highlighted is that lipid metabolism is different between the macrophage subsets, while M1 synthesize lipids in order to increase prostaglandin production, for instance, M2 macrophages metabolize lipids as a fuel for oxidative metabolism [65]. Therefore, the transcription factors that regulate the lipid metabolism, in special the PPAR family, are essential for lipid-handling in macrophages and support the alternative phenotype [66]. On the other hand, the oversupply of lipids to ATM found in WAT of obese subjects induces lipotoxicity and induction of pro-inflammatory profile and AET plays an essential role in the reduction of this lipid overload.

Despite the division of macrophages subsets into classical M1 and alternative M2, new approaches have allowed us to understand that ATM in obese mice is more complex and it presents an intermediate phenotype. In a recent review Caslin et al. discuss the complex metabolism and expression of inflammatory mediators and metabolic routes in ATM of obese and lean mice [65]. Obesity causes recruitment of not only macrophages but other immune cells to WAT such as neutrophils and lymphocytes that are responsible for sustaining the inflammation too. Moreover, like in macrophages, 16 weeks of AET mitigates neutrophil and CD8 lymphocytes infiltration in WAT from obese mice [67,68].

In summary, it is well-established that AET induces an anti-inflammatory milieu associated with metabolic benefits in WAT of obese mice which is related to alternative activation of immune cells resident in WAT. However, the mechanisms that regulate the metabolic pathways and the association with modification on inflammatory activations in adipose tissue-associated immune cells, still remain unclear. The disclosure of these pathways would be helpful in the treatment of several diseases linked to low grade inflammation.

### 4.2. Studies in Humans

Data on the isolated effects of AET in humans is more scarce due to the existence of a variety of protocols for weight loss that are associated with deeper lifestyle changes such as caloric restriction combined with exercise. Moreover, many studies are based on combined exercise protocols (resistance and aerobic exercise).

Twelve weeks of combined training induced the modification of seven transcripts in subcutaneous WAT from normoglycaemic and eutrophic sedentary subjects, while the same protocol induced modificationsin the expression of 90 transcripts in SAT of dysglycaemic and overweight sedentary men [69]. Another study with the same period of intervention (12 weeks) of combined exercise showed that insulin sensitivity and cardiorespiratory fitness were improved by exercise training. Moreover, the lipid peroxidation was reduced and mRNA expression of antioxidant enzymes was increased in SAT from the gluteal region of obese black African women [70]. Interestingly, the difference on transcriptome of abdominal and gluteal SAT was impacted by the combined training in the same population. At baseline, the difference between the two depots of SAT was limited to 15 transcripts but rose to 315 after exercise intervention [71], indicating that exercise induces to a higher heterogeneity between the WAT depots.

In regarding to weight loss, the European Guidelines for Obesity Management in Adults support that the notion that long-term AET shows efficacy in reduction of body adipose mass only [72]. As aforementioned the maximal rate of fatty acid oxidation is found in aerobic exercise at 60–65% of VO_2_ max [73]. Therefore, lipolysis is an essential mechanism to maintain the energetic supply (fatty acids) for skeletal muscle contraction. Verboven et al. showed that the lipolysis of SAT is altered by obesity and insulin sensitivity after 12 weeks of exercise training. The contribution of adrenergic stimulated lipolysis is higher in insulin sensitive obese individuals compared with non-insulin sensitive obese subjects or insulin-sensitive lean subjects. Moreover, the authors showed that the participation of adrenergic lipolysis represents approximately 40% of the total lipolysis ratio in insulin-sensitive obese subjects and it is abrogated in insulin-resistant obese ones [74].

Many proteins are candidates for stimulating adrenaline-independent lipolysis. IL-6 is the most studied cytokine in the context of AET. This cytokine is a pleiotropic factor that regulates many immunometabolic tasks, from the acute phase of immune response [75] to glucose and lipid metabolism [76,77]. In the late 90sPedersen‘s group showed that aerobic exercise induces to a huge IL-6 serum concentration released mainly by skeletal muscle [78]. Recently two very elegant studies showed the role of IL-6 released into circulation during the AET in promoting lipolysis in visceral, pericardial and epicardial WAT [79,80]. Thus, obese humans submitted to12 weeks of AET with a group that received tocilizumab (IL-6 receptor antibody) to block the IL-6 signaling confirmed that visceral, pericardial and epicardial weight loss in obese subjects were dependent on IL-6 [79,80].

In summary the study of immunometabolism in obese humans after AET are scarcer than animal models. Thus, many molecular studies need to be done to expand our knowledge in this area. It is unquestionable that AET is the main intervention for reducing visceral adipose tissue, systemic inflammation, improves cardiorespiratory fitness, and insulin sensitivity compared to other long-term pharmacological interventions as reviewed in a recent meta-analysis [81].

## 5. Adipose Tissue and Cancer: Linked by Inflammation

After cardiovascular diseases cancer is the second leading cause of death worldwide. Moreover one third of these deaths could be prevented by modifying life habits such as elevated tobacco usage, alcohol consumption and higher body mass index which is closely related to healthy diet and physical inactivity [82]. Despite the fact cancer etiology is not exclusively lifestyle-dependant there are somatic mutations involved which are triggered by errors on DNA replication or on its repair machinery. These mutations are more susceptible in some types of cancer such as melanoma and lung cancer, which may be caused by ultraviolet light exposure and tobacco smoking, respectively [83]. Although, among other causes that have been associated to cancer, obesity is one of the most concerning ones. Studies have shown that approximately 20% of all cancers with especial focus on breast [84], ovarian [85], endometrial [86], prostate [87], colon [88], and pancreatic [89] are strongly related to an excessive body mass index [90].

Given the complex biology of cancer, a malignant tumour should not be seem as just an isolated group of cells withfast growth, resistance to death and tendency to evadeother tissues. Surrounding and inside a tumour there are a variety of cells coexisting and/or supporting it. The tumour microenvironment must not be neglected and understanding how it is formed or how cells are recruited is essential for developing therapies [91]. Tumour characteristics have been elucidated, however its metabolic reprogramming or its ability to modulate the immune system in order to prevent self-destruction or even orchestrate an inflammatory tumour-promoting environment are features that still need clarification [92].

The origin of the inflammatory cells present in tumour stroma is based not only on fully differentiated immune cells from bone marrow but also from immature myeloid progenitors, identified as myeloid-derived suppressor cells (MDSC) that are stimulated to a pro-tumour activity [93]. Furthermore, immune cells presenting anti-tumour activity are also found in the tumour microenvironment, resulting in an antagonistic network that produces and secretescytokines and growth factors. So, what makes the tumour overwhelm the immune system and attract excessive pro-tumour cells? This answer can be in the chronic low-grade inflammation, as previously mentioned, a state triggered by hypertrophy of VAT and lack of physical activity that has been associated with cancer initiation, promotion and progression [94].

Tumours takes advantage of the host not only by modulating the immune system to promote a suitable environment based on inflammation but also use adipose tissue as a nutrient source that provides substrate for their replication and development. It is known that the Warburg effect is based on the glycolitic metabolism as a benefic way to cell anabolism [95] however some cancers have showing sustained fatty acid uptake from adipose tissue nearby and increased FA and cholesterol synthesis by de novo FA synthesis [96].

Moreover, fatty acids metabolism in tumour cells can be rerouted to prostaglandin synthesis pathways. It is well established that the function of prostaglandin (PG) E2 (especially, but many other inflammatory lipid mediators too) are associated with sustained tumour growth [97]. PGE 2 is increased in breast cancer, glioblastoma, colorectal cancer, urothelial carcinoma, among others [98,99,100,101], confirming that PGE2 is essential for tumour proliferation signaling [97], and it is necessary to sustained alternative macrophage polarization [100].

Poor cancer prognosis is linked to obesity-related chronic inflammation, abnormal adipokine secretion and browning [102]. When a tumour grows, it occurs faster than non-cancer cells, so usually this cell mass is poor vascularized and a hypoxic microenvironment results, leading to extracellular matrix remodeling [103]. The hypoxia will force the cells to adapt to a more glycolytic metabolism [104,105] while the extra cellular matrix modifications will favour metastasis [103]. Unfortunately, tumour cells are not alone on this metabolic switch; macrophages are also affected by the increased production of lactate, nitric oxide, reactive oxygen species and prostaglandins thus they undergo to a more glycolitic metabolic reprogramming [106].

Tumour-associated macrophages (TAMs) represent the majority of the innate immune cell population in the tumour microenvironment and they are deeply involved in cancer-related inflammation so they are potential targets for cancer treatment [107]. The M1 subtype is related to the Th1 response (production of pro-inflammatory cytokines and reactive oxygen/nitrogen species) acting as a host defense and consequently favouring a good prognosis by acting against tumours [108]. On the contrary, M2 subtype polarization is known to be tumour-promoting due to its role in angiogenesis and matrix remodeling factors that facilitate tissue evasion and metastasis [109]. Despite this duality, the tumour is able to favour its own progression by stimulating pro-tumorigenic factors in order to recruit TAMs and secrete growth factors [110]. Studies showed that the MSDC were able to oblate T cell via arginase secretion [111] and the production of IL-10 and TGF-β [112] resulted in suppression of adaptive immunity.

Macrophages are not the only ones that may undergo polarization in a tumour microenvironment, thus neutrophills, which are traditionally involved in defense against infection, can present anti-tumorigenic (N1) and pro-tumorigenic (N2) subpopulations and the recruitment of N2 neutrophills have been related to a cancer-associated fibroblasts production of TGF-β [113]. Furthermore, NK cells, like CD8+ T cells, act against the tumour by inducing pro-apoptotic and pro-inflammatory factors such as TNF-α, IL-6, IFN-γ and granulocyte-macrophage colony-stimulating factor (GM-CSF) in an attempt to induce proliferation and differentiation of hematopoietic cells [114].

In this context, adipose tissue has a valuable role in cancer development and progression by collaborating in obesity-related insulin resistance, inflammation and adipokine production that feed the loop: immune cells and adipocytes. In the case of breast cancer more specifically, the mechanisms that have been purposed as key factors in this interaction include leptin, adipose tissue inflammation, insulin and insulin growth factor(IGF-1) and sex hormones [115].

Leptin is secreted by adipose tissue and its main function is to regulate energy expenditure and food intake by acting in the central nervous system and peripherally (liver, adipose tissue and skeletal muscle). Obese individuals, despite higher levels of leptin, present limitations in its transport and signalling [116]. The mechanism that may explain the leptin receptor inhibition is mediated by SOCS3 (suppressor of cytokine signalling 3) whose Tyr985/SOCS3 interaction prevents leptin binding to its receptor [117]. Leptin and leptin receptor are found in epithelial cancer cells and they are linked to tumour growth, cell death impairments and angiogenesis induction via increased FGF-2 and VEGF expression [118], moreover leptin has been shown to be related to chemotherapy responsiveness [119]. Studies have suggested that leptin levels can be a good predictor for breast cancer diagnosis [120], prognosis [121] and survival [122]. Additionally, leptin and leptin receptor-deficient animal models presented resistance to induced tumor development, corroborating the important role of leptin in tumorigenesis [123,124].

Adiponectin is another hormone secreted by adipose tissue which is important for insulin sensitivity [125] via AMPK activity enhancement [126]. AMPK contributes to insulin sensitivity due to an increase in glucose uptake in insulin sensitive organs, but, unlike leptin, adiponectin is reduced in obese subjects, thus its anti-inflammatory, anti-proliferative and pro-apoptotic effects are compromised in this population [127]. Solid tumours present adiponectin receptors [128] and studies with MCF-7 human breast cancer have indicated positive apoptotic response effects via p53 are modulated by AMPK phosphorylation induced by adiponectin [129,130] specially in ERα negative cancer [130,131,132]. Besides that, adiponectin is able to inhibit the activity of an enzyme responsible for estrogen production, aromatase, one of the major risk factors for breast cancer development in postmenopausal women [90].

IGF-1 also play an important role on cancer, especially in obesity, a condition in which IGF-1 concentration in serum is elevated in a long-term manner, thus its effects are more related to growth factors, such as anti-apoptotic and mitogenic properties [133]. Besides, evidence has shown a link between IGF receptor activation and the development of breast cancer together with the fact that this receptor overactivation is associated to resistance to radio therapies and tumor recurrence [134].

Adipose tissue plasticity is really important in this cancer crosstalk. It is clear that WAT is an energy supply, has angiogenic effects and collaborates in inflammation related to cancer [135]. Pink adipocytes, localized on mammary glands, may also impact on immune cells modulation and empower breast cancer development. Pink AT undergoes reversible transdifferentiation after lactation and PPARγ plays an important role in this process [136]. In colorectal cancer C/EBP-α, PGC-1α and NF-kappaB has been described as potential biomarkers [137]. Brown adipose tissue volume is associated with tumour recurrence/mortality, regardless of age, body mass index, gender or tumour type [138]. Cancer cachexia treatments consist of pharmacological and non-pharmacological strategies including the activation of PPARγ via improvements on insulin sensitivity and adipose tissue loss [139] and attenuation of inflammation by exercise, specially the aerobic training, which represents a safe and low cost intervention [140].

AET has been shown to be an important tool in cancer patient treatment based on controlling inflammation related to obesity and cancer cachexia [141], but also presenting positive effects on cancer-related fatigue [142,143], a subjective sense of physical, emotional, and/or cognitive tiredness extensively reported by cancer patients during and after treatments. The role of AET in cancer adipose tissue and inflammation will be further discussed below.

## 6. Therapeutic Effects of Exercise in Cancer

Exercised-related protective effects and benefits were firstly reported in cancer patients in the 80s [144,145]. Since then the number of studies in this field rose substantially and exercise started to be considered a tool for the treatment for cancer-related fatigue [146], for functional, psychological and emotional well-being [147] and for cancer cachexia [148]. Recently, it has been suggested that exercise is a worthy therapy in many cancer treatments at all stages [149], along with radiotherapy, chemotherapies and immunotherapies [150], including for cancer-related cognitive impairment [151].

Despite the lack of clinical trials considering different exercise types, intensity and frequency, many other variables have to be considered such as cancer types, stages, and therapies applied. Thus, identifying and advocating a general guideline for exercise training in cancer patients is an extremely difficult mission. Nonetheless there is no doubt that exercise is safe and efficient during and after cancer treatment [152]. In this sense and due to the higher prevalence of studies based on aerobic exercise, this review will focus only on this exercise model and its effects on adipose tissue and/or immune responses. 

### 6.1. Studies in Animals

Firstly, it is interesting to highlight the benefits of exercise in cancer prevention in a lifelong manner, once the considerable impact that aging has especially on immune system and inflammation [153]. A recent study conducted in a naturally aging mouse model, systemic inflammation and cytokines associated with cancer progression were mitigated by lifelong lasting aerobic exercise. Furthermore, the presence of malignant tumours was only observed in sedentary mice [154]. Goh and collaborators also reported the effectiveness of exercise dose-response in cancer progression. Mice that had run longer distances prior to cancer injection (at 18 months old) developed smaller tumours compared to the shorter distance runners or sedentary animals [155]. Another interesting study reported that the protective effect of low-intensity exercise prior to cancer establishment in delaying growth and reducing breast tumor volume was mediated by a reduction of M2 macrophage infiltration [156].

In a spontaneous model of breast cancer, a vigorous exercise training protocol of 20 weeks (60 min/d, 20 m/min, 5% inclination and 6d/wk) was able to reduce the volume and number of tumours (metastasis) accompanied by decreased plasma concentration of MCP-1 and IL-6 [157]. Despite the high intensity/volume protocol, it may be explained by an accumulated protective effect of exercise and also by the importance of attenuated systemic inflammation for reducing cancer risk.

Reducing both systemic and tumour microenvironment inflammation are extremely important for lower cancer risk and development, although cell cycle arrest is another essential aspect that can be mediated by exercise. Yu and collaborators tested a 20 m/min, 60 min/day and 5 days/week for 10 weeks protocol in mouse skin cancer model and found that exercised mice showed reduced IGF-1 bioavailability via IGFBP-3 augmented expression and IGF-1 downregulation via PTEN (a tumour suppressor gene) overexpression [158].

Due to the elevated cancer prevalence in obese individuals, obese mice models are quite common in the literature. In a very elegant study Theriau et al. used serum-enriched culture media from high fatty diet sedentary and voluntary exercised mice in breast cancer cells. The authors concluded that exercise, in a dose-dependent manner, abolished the deleterious effects induced by obesity and also inferred that this positive outcome was modulated by the adipokine secretion (adiponectin and leptin) profile and adiponectin receptor 1 enhanced expression, that in turn, act on cell cycle inhibition [159]. Another study evaluated the cessation of a high fat diet, followed or not by exercise, on colon rectal cancer risk and the authors concluded that the positive effect of exercise was mediated by lowering inflammation to a similar level found on control animals that have never been obese before [160].

It was previously mentioned that tumour cells can increase production and recruitment of MDSCs from the bone marrow via granulocyte-macrophage colony-stimulating factor (GM-CSF). However, exercise was able to delay this recruitment and consequently enhance immunotherapies based on checkpoint inhibitors [161]. Additionally, exercise alone (by treadmill running at 18 m/min, 30 min per session, 5 days/week for 4 weeks) delayed triple negative breast cancer tumour growth and favored intratumoral leukocyte CD8+ T cells activation, demonstrated by elevating expression of CD69 and reducing frequency of MDSCs. Besides, when associated to radiotherapy and immunotherapy there was a tumoral mitigation of MDSCs accumulation which contributes to NK cell activation (anti-tumour activity) [162].

With regards to cytokines and growth factor concentration, an exercise dose-response was tested (6, 10 and 15 m/min) in a short-term protocol (20 days) and for this period the higher the intensity the more effective in delaying tumour growth. This outcome was explained by mobilization and redistribution of NK cells into the solid tumour via increased adrenaline and IL-6 in serum [163]. While epinephrine mobilizes NK cells into the circulation, muscle-derived IL-6 is responsible for NK cell redistribution [164]. These studies may explain the greatest efficacy of highest exercise intensity since hormone and IL-6 production are positively correlated to exercise intensity [165,166]. Paradoxally, lower concentrations on intra tumoral IL-6 showed a positive anti-tumour effect in association with VEGF reduction resulting in less tumour volume in groups that did exercise before and after malignancy [167]. It has been demonstrated that IL-6 is a potential biomarker in colorectal cancer patients and in cancer cell lines also associated with likelihood of cancer recurrence [168] and cachexia [169]. Other cytokines related to cancer-mediated inflammatory process such as IL-8, TNF-α and CRP were also reduced by exercise training (18 m/min for 12 weeks, at 30 min for 5 days per week) in breast cancer mice [170].

It is worth remembering that the group of cytokines regulated by exercise in tumour bearing mice is basically the same of those ones present in obese cancer patients. However, a weakness in the animal studies is the lack of adipose tissue in cancer models for analysis. Indeed, visceral obesity, considering the adipocytes and infiltrated immune cells, is particularly related to the increase concentration of this cytokines (TNF-α, IL-6, adiponectin, visfatin and so on), evidencing the important link between adipose tissue, inflammation and cancer. 

### 6.2. Studies in Humans

It is a consensus that exercise can bring numerous benefits to cancer patients, whether they are already practicing activities before the tumour emergence or if started after diagnosis [171]. A review published in 2008 by the *British Journal of Sports Medicine* concluded that there is a 25% reduction in the risk of breast cancer in groups of more physically active women [172]. Some direct and indirect mechanisms by which exercise affects cancer are further discussed by Thomas, however we would like to explore those somehow connected to obesity such as energy metabolism and insulin resistance, leptin, IGF-1, chronic inflammation and obesity-related hormones and cytokines [173].

Growth hormones and IGF are mitogenic factors also produced in acute exercise sessions, in this sense, the hypothesis that various bouts of exercise could lead to tumour malignancy was tested by Rundqvist and colleagues in experiments that exposed prostate cancer cell lines to serum of athletes pre- and post-exercise session. Cells incubated with post-exercise serum presented 31% growth inhibition and further serum analysis showed that the inhibitory effect was possibly mediated by increased levels of IGFBP-1 [174]. In a human model of breast cancer survivors that performed exercise on cycle ergometers three times per week for 15 weeks, no differences in insulin, glucose or IGFBP-1 were observed, although IGF-1 was reduced and IGFBP-3 was elevated in the exercise group [175]. Both IGF binding proteins IGFBP-1 and 3 are inhibitors of insulin growth factor and are associated with lower risk of certain types of cancer such as prostate [176,177], colorectal [178], esophageal [179] and brain tumours [180]. Diet can affect IGF production thus it can be considered an important factor linking poor nutrition with cancer.

Regards to energy metabolism, the hyperglycaemia and hyperinsulinemia observed in insulin resistant patients are also associated to elevated risk of cancer and a widely known mechanism of increasing insulin sensitivity and glucose uptake is upregulating AMPK activity [181]. It was demonstrated that in MCF-7 breast cancer cultured with serum from human pre adipocytes led to cell proliferation yet AMPK and metformin treatments recovered the adipokine imbalance caused by LPS stimulation [129]. These findings reassure the importance of AMPK activation on obesity-related energy disturb, insulin resistance and cancer growth. Furthermore, AMPK downregulation is observed in chemotherapy with drugs such as doxorubicin, thus exercise may be considered an adjuvant therapy for restoring energy balance [182].

Inflammation, systemically or locally, is another factor that is strongly modulated by exercise, especially in the cancer context. Studies using a combination of aerobic and resistance exercise (16 weeks) in obese postmenopausal breast cancer survivors observed not only cardiometabolic and body composition progress but also anti-inflammatory effects. Post-exercise protocol, adipose tissue macrophages from the participants had lower M1 and higher M2 macrophages compared to their sedentary pairs. Additionally, IL-6 and TNF-α secretion also diminished, confirming the improvement on AT inflammation. Despite the well known inflammatory tumour microenvironment in cancer patients and tendency to M2 macrophage polarization, these results suggest the importance of physical activity in chronic inflammation attenuation and in minimizing cancer recurrence [183]. Another study performed on a similar population concluded that exercise training had positive effects on C-reactive protein, a widely known cancer marker [184].

Compared to resistance exercise, aerobic exercise has demonstrated to be more compliant in cancer survivors and had presented more efficiency on body composition management, as weight gain is commonly observed after cancer treatment. It was shown by a long term protocol (12 months) in which the aerobic group displayed significant improvements in aerobic capacity and muscle strength, including body composition benefits [185]. The engagement and exercise constancy during treatment is extremely important for cancer-related fatigue and sarcopenia, which are highly prevalence among cancer survivors due to chemotherapy toxicity [186]. It is important to note that the cancer type and stage of intervention, if conducted in patients at risk (prior to cancer emergence), during cancer treatment or survivors (after treatment), will deeply impact patients’ metabolic and physiological status. Thus, identifying and understanding each stage of the disease and the many possibilities of exercise as an intervention are indispensable for interpreting the studies outcomes or standardizing a protocol. Recently, Segal and colleagues published a guideline for adult cancer patient survivors or in treatment in which frequency, duration and intensity of exercise are discussed and some essential points are verified: (a) exercise is safe in moderate quantity; (b) the aims are improvements on muscular and aerobic fitness; (c) clinicians should suggest 150min/week of aerobic moderate-intensity exercise three to five times per week plus resistance training twice a week (two sets of 8–10 repetitions for 8–10 muscle groups), additionally each session must include warm-up and cool-down; (d) pre-existing comorbities must be considered; (e) group and supervised activities are preferred [187].

In a randomized clinical trial, women with increased risk of breast cancer were submitted to moderate intensity aerobic exercise for 5 months and they were subdivided into low-dose and high-dose (150 or 300 min/wk) groups. Not surprisingly, body composition was modulated and visceral adipose tissue was correlated with changes in cancer risk [188]. A dose-response effect of exercise on adiponectin and leptin was also demonstrated, nonetheless the dose response was dependent on changes in body fat [189]. Estrogen and hormonally-sensitive breast tissue decreased linearly in low and right doses of exercise, besides, no differences on estrogen levels was seem in the intervention groups [190]. The WISER Sister Study found augmented proinflammatory biomarkers after moderate to vigorous aerobic exercise intervention that lasted five menstrual cycles of healthy premenopausal women. However, the authors hypothesized that in this specific population the exercise-mediated mechanisms for cancer prevention may not be related to inflammation [191]. In the same direction, visceral adipose tissue of patients in colon cancer stage I to III submitted to an aerobic exercise protocol showed reductions in a dose-response manner, pointing to exercise as a cancer risk reducer for disease recurrence among survivors [192].

Exercise has a significant effect on serum leptin of obese individuals [193] although leptin levels in cancer patients seem to be controversial. In men with colorectal cancer submitted to 8 weeks moderate intensity walking in three 45-min sessions in each week body fat percentage was decreased but plasma leptin concentration did not change significantly [194]. On the other hand, in overweight/obese triple-negative breast cancer survivors, moderate-intensity aerobic exercise (150 min per week, for 12 weeks), despite no changes on serum cytokines and adipokines, had positive effects on body composition. Still, serum leptin and adiponectin and their ratio were significantly correlated with body mass index in the intervention group, confirming that changes in leptin and adiponectin may reflect the changes in adiposity with exercise intervention [195].

So far, it seems that the positive effects of exercise on obesity-related parameters of cancer are based on body composition changes. Still, the immune response modifications observed as consequence of aerobic training are more evident on tissue-specific rather than peripheral blood. Certainly, more human studies aiming immune changes are required to answer deeper questions in this complex cancer-related inflammation in exercise field. Figure 1 presents a summary of the alterations on adipose tissue in obesity and cancer before and after aerobic exercise training.

## 7. Conclusions

Exercise is an excellent tool to reduce the risk and severity of many chronic diseases. Immunometabolic modifications caused by aerobic exercise training are efficient in reducing low grade inflammation and increasing the immune surveillance. One of the chronic adaptations caused by exercise is mediated by energetic stress which induces beneficial molecular adaptations in adipose tissue and immune cells. Thus, the aerobic exercise training is responsible for immunometabolic modifications that are able to mitigate the inflammation-related metabolic disarrangement observed in obese populations. Moreover, the widely known strong association between obesity and cancer may be prevented by aerobic exercise training. Finally, the aerobic exercise is a low-cost tool with excellent results in the prevention and co-treatment of obesity and cancer. Many favorable outcomes are provided by the aerobic exercise training ability on managing body composition and adipose tissue remodeling. 

## Figures and Tables

**Figure 1 ijms-21-09745-f001:**
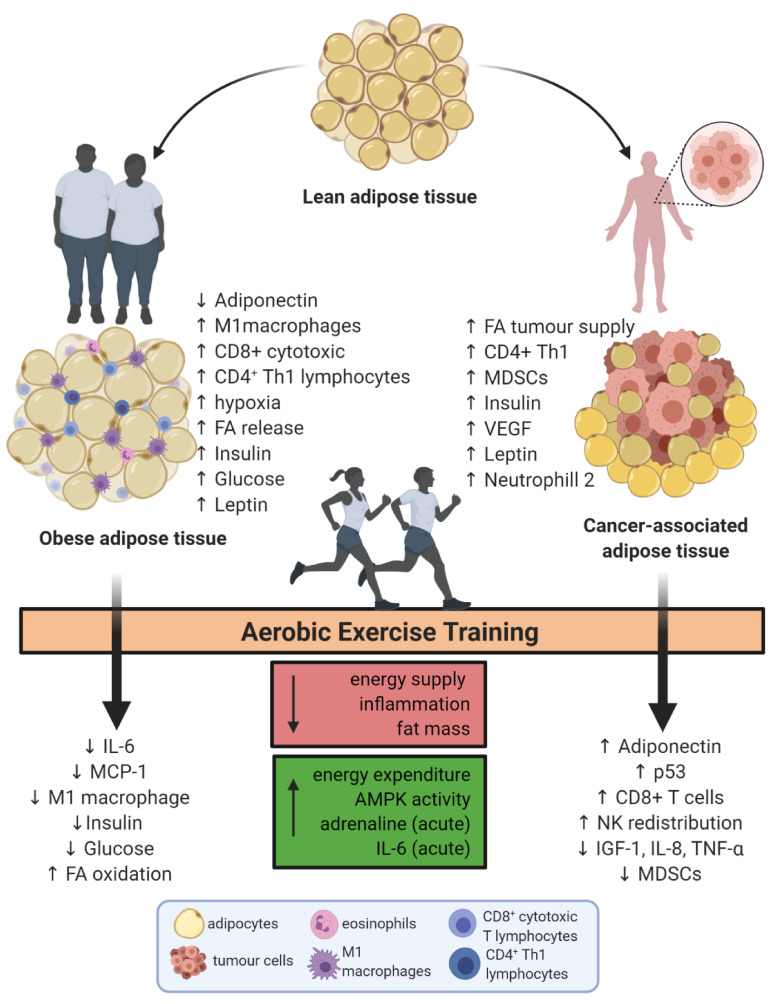
Aerobic exercise training effects on adipose tissue alterations related to obesity and cancer. IL-6 = interleukin 6; FA = fatty acids; MCP-1 = monocyte chemoattractant protein 1; NK = natural killer; AMPK = 5’ AMP-activated protein kinase; MDSCs = myeloid-derived suppressor cells; VEGF = vascular endothelial growth factor; IL-8 = Interleukin-8; TNF-α = tumour necrosis factor alpha. Created with BioRender.com.

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
