# Peer review of "Endurance Exercise Mitigates Immunometabolic Adipose Tissue Disturbances in Cancer and Obesity"

_ijms, 2020, doi:10.3390/ijms21249745_

Round 1

Reviewer 1 Report

The manuscript concerns a very interesting and current topic; adipose tissue performs immunoregulatory functions in addition to fat storage and the authors discuss the recent literature regarding the effects of exercise on the immunometabolism of adipose tissue. The conclusions are clearly and completely supported.

The authors concluded that aerobic exercise is an excellent therapeutic tool for control the inflammation of obesity, with important application in cancer prevention and care.

However, the review will benefit if the Authors take my suggestions into consideration:

1) I believe that review article is much stronger if are build on specific experimental evidence (by citing original articles) rather than on general concepts (eg. by citing reviews). I feel that this is particularly true for Authors that have extensive experience in a field.

This number must be drastically reduced and the author should support his statements by referring to the original research article presenting the data on which his statements are based (for example, line 57-58 references (8) and (9) are reviews articles that cited experimental data obtained from the same group (Morroni M et al, PNAS 2004; De Matteis et al, Stem Cell, 2009.)

2) The effects of exercise on fat are depot-specific. I  would like to suggest greater care in studying the effects of aerobic exercise on the immunometabolism of WAT, distinguishing those on the visceral (vWAT) from those on the subcutaneous (scWAT). You could also say something about the BAT.

Author Response

We agree with the critics and suggestions. 

We did substantial modification in the text in according with suggestions. 

The manuscript was improvement after review process.

Thank you for reviewer.

We attached the response letter.

Reviewer 1

The manuscript concerns a very interesting and current topic; adipose tissue performs immunoregulatory functions in addition to fat storage and the authors discuss the recent literature regarding the effects of exercise on the immunometabolism of adipose tissue. The conclusions are clearly and completely supported.

The authors concluded that aerobic exercise is an excellent therapeutic tool for control the inflammation of obesity, with important application in cancer prevention and care.

However, the review will benefit if the Authors take my suggestions into consideration:

1) I believe that review article is much stronger if are build on specific experimental evidence (by citing original articles) rather than on general concepts (eg. by citing reviews). I feel that this is particularly true for Authors that have extensive experience in a field.

This number must be drastically reduced and the author should support his statements by referring to the original research article presenting the data on which his statements are based (for example, line 57-58 references (8) and (9) are reviews articles that cited experimental data obtained from the same group (Morroni M et al, PNAS 2004; De Matteis et al, Stem Cell, 2009.)

Author’s comments: We agreed. We excluded some paragraphs to reach higher readability and replaced review references to original manuscripts, when it was possible.

2) The effects of exercise on fat are depot-specific. I would like to suggest greater care in studying the effects of aerobic exercise on the immunometabolism of WAT, distinguishing those on the visceral (vWAT) from those on the subcutaneous (scWAT). You could also say something about the BAT.

Author’s comments: We agree and inserted the text bellow: 

“The AET adaptations on BAT are still controversial. Less current studies showed that AET (swimming or running) in rodents increased the mitochondrial activity in BAT (Yoshioka, Yoshida et al. 1989; Oh-ishi, Kizaki et al. 1996). However, more recent research showed BAT mass is reduced after 8 weeks of AET (Wu, Bikopoulos et al. 2014). On the same direction, well trained athletes had decreased glucose uptake and cold-induced BAT activation compared with sedentary subjects (Vosselman, Hoeks et al. 2015). “

Reviewer 2 Report

This article aims to review the impact of exercise on obesity–associated cancers

Major Comments

We feel that the organization of the review is not well though. Some  chapters describing the pathophysiology are too long, and the parts relative to the effects of AET are  too short, except in the chapter about cancer.  However, in this chapter, there is a mixture of pathophysiology and exercise, which renders things difficult to understand. This chaper is also too long and must be reorganized.

For example, it is described in the pathophysiology part about cancer  and adipose tissue that M2 are of poor prognosis, but later on, the authors indicate that M2 macrophages  increase with AET, whitout discussing this point. Inversely p53 , or IGF1 have not been adressed in the first part, and they are present in the second part. We are also surprised to observe  that almost nothing is written about leptin, whereas this is one of the most important effects of AET on cancer ( Ando et al, for review, Cancers 2019). We feel that the authors have developed some pathophysiological events not linked to obesity . as for example in the paragraph about cytolytic functions. They should only focus on mechanisms linking excess Adiposity and Carcinogenesis Promotion, and indicate the effects of AET on those mechanisms.

Minor Comments to correct

In the introduction, the authors mention the existence of brown A adipocytes and white Adipocytes.  But  they should also mention that among white adipose tissue ( AT), AT sc is protective while TAV induces insulinresistance , as nicely demonstrated by Tran et al, Cell Metab. 2008.

Page 2, line 69 : The authors indicate that the infiltration of macrophages precedes that of T cells in obese AT, but they should also mention the very smart study from Nishimura et al    who  demonstrated the inverse  situation. Indeed, they have shown that CD8+ T cells preceed macrophage infiltration in obese adipose  tissues, and subsequently  induce differentiation of monocytes into M1 macrophages ( Nishimura et al, Nat Med 2009)

Moreover, it should be interesting to add in this description  that  Th2 macrophages are likely to protect against insulin-resistance as demonstrated by Winer  et al ( Nat Med, 2009) . While Th1 macrophages contribute to insulin resistance. Same for M2 and M1 macrophages ( Lumend, J. Clin Invest. 2007). However this latter dogma is a controversal issue at present, since Q. Jya, reported  that adipose-tissue macrophages  may primarily play a role in tissue remodeling rather than in metabolic pathology. (Q Jya, et al, Am J Physiol Endocrinol Metab, 2020). This can be added as well

Page 4, last paragraph : PPARg is a transcription factor….M2 polarization. The reference is missing

Page 5 : Remodeling obese   AT by AET . This chapter describes the mechanisms of AT expansion, but very poorly the effects of AET. This chapter is not well equilibrated

Page 6 : it is mentioned in line 219, that adiponectin protein expression is decreased in adipocyte. which is striking. Indeed most papers have indicated an increase of serum  adiponectin and a decrease of leptin by AET ( Simpson et al, Obesity 2008, Yu et al, Horm Metab Res, 2017).The authors should mention it, and explain their discordant results.  In this paragraph,  the authors did not add any comment on the effects of leptin , which is clearly missing; leptin has been very poorly mentioned in their paper.

In page 9, last paragraph, the description of leptin effects on satiety is confusing. Indeed, the sentence about the blood brain barrier is difficult to understand , because leptin is known to access to the hypothalamus, but its transport can be decreased by obesity ( Jecquier et al, Ann N Y Acad Sci, 2002)) This should clearly indicate it  like this . However the most important effects of hyperleptinemia is increased resistance of its receprors to leptin , due to increased transcription of SOCS3, a target of leptin-mediated STAT-3 activation. SOCS3 mediates a feedback inhibition of the leptin receptor (Bjourbaeck et al, J. Biol Chem, 2000).  The authors should erase their sentence from  « Obese individuals….SOCS3, because it is very confusing and replace it.

Page 7, line 243. In this chapter the authors indicate that M1 use the lipid synthesis pathways , and M2 use lipids as fuel to oxidative metabolism. This sentence is not clear. . It is  usually reported that M1 use  glycolysis , while M2 use oxidative phosphotylation ( Caslin, Immunol Rev 2020)). Moreover, the authors inversed reference 64 with 65.

Page 8 , line 301. The authors should replace « contitionated to action of IL-6 » by dependent on IL-6

Page 9, line 363,. « M2 subtype polarization  is known to be tumor promoting » the reference is missing

page 12, line 519 : The authors indicate that AMPK downregulation is observed by chemotherapy, whereas they mentioned above that hyperinsulinemia increases the risk of cancer via downregulation of AMPK. Therefore, this seems discrepant.  A discussion about this issue is recommended , as well as a reference about the chemotherappy effects on AMPK down.

Finally , the authors  indicate that exercise  may restore the energy balance. Is it by increasing AMPK activity ?  If this is the case , a reference should be added . If not, this should not be mentioned herein.

Line 525, the authors indicate that AET-mediated lower M1, but higher M2macrophages was observed in obese breast cancer patients, as compared with sedentary pairs. However,  this is discordant with the above mentioned pathogenic  effects of M2 macrophages on cancer ( page 9, line 363). Therefore, these discordant results need to be discussed  

Author Response

Thank you so much for your review. 

The critics and suggestion were very important to improve our manuscript.

In text we did the substantial review to improve the readibility and modified as your suggestions, that we are in agree. 

Our response follow bellow and in attachment.

Reviewer 2

Major Comments

We feel that the organization of the review is not well though. Some chapters describing the pathophysiology are too long, and the parts relative to the effects of AET are too short, except in the chapter about cancer.  However, in this chapter, there is a mixture of pathophysiology and exercise, which renders things difficult to understand. This chapter is also too long and must be reorganized.

Author’s comments: In the manuscript draft the lengths of the chapters were not taken into consideration. The author’s guideline indicated no limit of word or pages. However we agreed with the reviewer and some parts of obesity pathophysiology were excluded. Regards to cancer topic, it was subdivided in “5. Adipose tissue and cancer: linked by inflammation” (containing pathophysiology) and “6. Therapeutic effects of exercise on cancer” (exercise effects) thus we hope that the reading is clearer.

For example, it is described in the pathophysiology part about cancer and adipose tissue that M2 are of poor prognosis, but later on, the authors indicate that M2 macrophages  increase with AET, without discussing this point.

Author’s comments: This point was further commented.

Inversely p53, or IGF1 have not been addressed in the first part, and they are present in the second part.

Author’s comments: Actually, p53 was briefly comment in topic 5 (pathophysiology) “Solid tumours present adiponectin receptors (Katira and Tan 2015) and studies with MCF-7 human breast cancer have indicating that positive effects on apoptotic response via p53 are modulated by AMPK phosphorylation induced by adiponectin (Grisouard, Dembinski et al. 2011; Shrestha, Nepal et al. 2016) specially on ERα negative cancer (Shrestha, Nepal et al. 2016; Mauro, Naimo et al. 2018; Naimo, Gelsomino et al. 2020).” however it was deleted from the topic 6 (exercise effects) once it is not related to adipose tissue mediated inflammation in cancer.

IGF1 comments were added in the first part:

“The IGF-1 also play an important role on cancer, especially in obesity, a condition in which IGF-1 concentration on serum are elevated in a long-term manner, thus its effects are more related to growth factors, such as anti-apoptotic and mitogenic properties (AsghariHanjani and Vafa 2019). Besides, evidences have shown a link between IGF receptor activation and development of breast cancer together with the fact that this receptor overactivation is associated to resistance to radio therapies and tumor recurrence (Surmacz 2000).”

We are also surprised to observe that almost nothing is written about leptin, whereas this is one of the most important effects of AET on cancer (Ando et al, for review, Cancers 2019).

Author’s comments: More information was added regards to leptin. The following reference was included: (Ando, Gelsomino et al. 2019).

We feel that the authors have developed some pathophysiological events not linked to obesity, as for example in the paragraph about cytolytic functions. They should only focus on mechanisms linking excess Adiposity and Carcinogenesis Promotion, and indicate the effects of AET on those mechanisms.

Author’s comments: some descriptions not linked to excess adiposity were excluded.

Minor Comments to correct

In the introduction, the authors mention the existence of brown adipocytes and white Adipocytes.  But they should also mention that among white adipose tissue (AT), AT sc is protective while TAV induces insulinresistance, as nicely demonstrated by Tran et al, Cell Metab. 2008.

Author’s comments: Thank you for this suggestion. We inserted this paper and this context in the paragraph.

“While the increase on the VAT depot is related with low chronic inflammation and insulin resistance, the SAT is associated with improvement on glucose homeostasis and it is not linked with chronic inflammation. In fact, the transplantation of SAT to VAT depots, in mice, improved the glucose homeostasis and decreased body fat mass [3].”

Page 2, line 69: The authors indicate that the infiltration of macrophages precedes that of T cells in obese AT, but they should also mention the very smart study from Nishimura et al   who demonstrated the inverse  situation. Indeed, they have shown that CD8+ T cells precede macrophage infiltration in obese adipose tissues, and subsequently induce differentiation of monocytes into M1 macrophages (Nishimura et al, Nat Med 2009)

Author’s comments: Thank you, we inserted this:

“While Talukdar and colleagues (2012) showed that the first immune cells recruited are neutrophils, which in turns, increases chemoattraction of macrophages (Talukdar, Oh et al. 2012), Nishimura and colleagues (2009) showed that the first immune cell infiltrated in inflammation is CD8 T lymphocytes with reduction on T regulatory CD4 lymphocytes and a posteriori recruitment of macrophages (Nishimura, Manabe et al. 2009).”

Moreover, it should be interesting to add in this description that Th2 macrophages are likely to protect against insulin-resistance as demonstrated by Winer et al ( Nat Med, 2009) . While Th1 macrophages contribute to insulin resistance. Same for M2 and M1 macrophages (Lumend, J. Clin Invest. 2007). However this latter dogma is a controversal issue at present, since Q. Jya, reported  that adipose-tissue macrophages  may primarily play a role in tissue remodeling rather than in metabolic pathology. (Q Jya, et al, Am J Physiol Endocrinol Metab, 2020). This can be added as well

Author’s comments: This part was improved as suggested:

“Therefore, the balance between Th1, T regulatory and Th2 lymphocytes (Winer, Chan et al. 2009) and M1 and M2 macrophages (Lumeng, Bodzin et al. 2007)  is essential to homeostasis of AT. An elegant study demonstrated that metabolic disturbances induced by elevated body mass index are associated with insulin resistance, inflammation and dyslipidemia. Whereas the M1:M2 ratio is not a predictor, howsoever the role of macrophages subset may be important to remodeling of AT (Jia, Morgan-Bathke et al. 2020).”

Page 4, last paragraph: PPARg is a transcription factor….M2 polarization. The reference is missing

Author’s comments: Reference was placed.

Page 5: Remodeling obese   AT by AET. This chapter describes the mechanisms of AT expansion, but very poorly the effects of AET. This chapter is not well equilibrated

Author’s comments: We agree. We excluded the pathophysiological paragraphs and included the role of lipids remodeling induced by exercise:

“Moreover, the AET is able to induce the remodeling of lipid species in SAT and BAT. Phospholipids and triacylglycerol were altered after three week of voluntary running. This interesting find showed that AET promotes a modification on the pattern of lipids and it is dependent on adipose tissue depot (May, Baer et al. 2017). However, the physiological effects of these changes on lipid species should be better investigated to understand the function of each lipid class in the adipose tissue remodeling by AET.”

Page 6: it is mentioned in line 219, that adiponectin protein expression is decreased in adipocyte. Which is striking. Indeed most papers have indicated an increase of serum adiponectin and a decrease of leptin by AET (Simpson et al, Obesity 2008, Yu et al, Horm Metab Res, 2017). The authors should mention it, and explain their discordant results.  In this paragraph, the authors did not add any comment on the effects of leptin, which is clearly missing; leptin has been very poorly mentioned in their paper.

Author’s comments: Very useful suggestion of paper (Simpson and Singh 2008), thank you. 

“Moreover, IL-6 and adiponectin protein expression were attenuated in isolated adipocytes but not on AT (stromal plus adipocytes). Thus, the contribution of adipocytokines in obesity and the AET effects seems to be different between WAT depots and isolated adipocytes (Silveira, Biondo et al. 2020). In a study that fractionated adipocyte size and evaluated cytokines secretion, concluded that the very large adipocytes are the responsible for promoting inflammation (Skurk, Alberti-Huber et al. 2007), suggesting that VAT and SAT may play different roles on the adipocytokines secretion and consequently on local inflammation. Additionally, chronic diseases and diagnosed pathologies are able to further reduce adiponectin levels and exercise does not seem to be an effective tool for restore adiponectin levels howsoever the body composition changes mediated by exercise programs may be the most important outcome for adiponectin modulation (Simpson and Singh 2008).”

A paragraph for exercise-mediated leptin effects was included:

“Exercise has a significant effect on serum leptin of obese individuals (Yu, Ruan et al. 2017) although leptin levels on cancer patients seem to be controversial. In men with colorectal cancer submitted to 8 weeks moderate intensity walking in three 45-min sessions in each week body fat percentage was decreased but plasma leptin concentration did not change significantly (Nuri, Moghaddasi et al. 2016). On the other hand, in overweight/obese triple-negative breast cancer survivors, moderate-intensity aerobic exercise (150 min per week, for 12 weeks), despite of no changes on serum cytokines and adipokines, had positive effects on body composition. Still, serum leptin and adiponectin and their ratio were significantly correlated with body mass index in the intervention group, confirming that changes in leptin and adiponectin may reflect the changes in adiposity with exercise intervention (Swisher, Abraham et al. 2015).”

In page 9, last paragraph, the description of leptin effects on satiety is confusing. Indeed, the sentence about the blood brain barrier is difficult to understand , because leptin is known to access to the hypothalamus, but its transport can be decreased by obesity ( Jecquier et al, Ann N Y Acad Sci, 2002)) This should clearly indicate it  like this . However the most important effects of hyperleptinemia is increased resistance of its receprors to leptin, due to increased transcription of SOCS3, a target of leptin-mediated STAT-3 activation. SOCS3 mediates a feedback inhibition of the leptin receptor (Bjourbaeck et al, J. Biol Chem, 2000).  The authors should erase their sentence from  « Obese individuals….SOCS3, because it is very confusing and replace it.

Author’s comments: We agreed in the confusing manner it was written, so a new paragraph was done.

 “Obese individuals, despite of higher levels of leptin, present limitations on its transport and signalling (Jequier 2002). The mechanism that may explain the leptin receptor inhibition is mediated by SOCS3 (suppressor of cytokine signalling 3) whose interaction Tyr985/SOCS3 prevents leptin binding to its receptor (Bjorbak, Lavery et al. 2000). “

Page 7, line 243. In this chapter the authors indicate that M1 use the lipid synthesis pathways, and M2 use lipids as fuel to oxidative metabolism. This sentence is not clear. . It is usually reported that M1 use  glycolysis , while M2 use oxidative phosphotylation ( Caslin, Immunol Rev 2020)). Moreover, the authors inversed reference 64 with 65.

Author’s comments: Here we replaced the reference. I believe that here the two information are different. Firstly it was explained that the M1 is more glycolytic and M2 more oxidative. But, in the referred part we intend to say that the lipids in M2  are oxidized by energy generation, and in the M1 the lipids are synthesized from glucose by lipogenesis de novo.  For example, M1 macrophages synthesize lipids and produce inflammatory lipid mediators like eicosanoids while M2 macrophages tend to take up and oxidize lipids. This relationship appears in the context of obesity as well.

“An important point to be highlighted is that lipid metabolism is different between the macrophage subsets, while M1 synthesize lipids in order to increase prostaglandin production, for instance, M2 macrophages metabolize lipids as fuel to oxidative metabolism (Caslin, Bhanot et al. 2020)”

Page 8 , line 301. The authors should replace « contitionated to action of IL-6 » by dependent on IL-6

Author’s comments: We changed this.

Page 9, line 363,. « M2 subtype polarization  is known to be tumor promoting » the reference is missing

Author’s comments: (Chen, Song et al. 2019) was added

page 12, line 519 : The authors indicate that AMPK downregulation is observed by chemotherapy, whereas they mentioned above that hyperinsulinemia increases the risk of cancer via downregulation of AMPK. Therefore, this seems discrepant.  A discussion about this issue is recommended , as well as a reference about the chemotherappy effects on AMPK down.

Finally, the authors  indicate that exercise  may restore the energy balance. Is it by increasing AMPK activity?  If this is the case , a reference should be added . If not, this should not be mentioned herein.

Author’s comments: Yes, AMPK activation is considered a good prognosis on cancer treatments (Zulato, Bergamo et al. 2014) however we did not intend to mention that high level of insulin leads to cancer via AMPK downregulation. What we tried to say was that AMPK activation could be a good way of restoring insulin and glucose levels.Thus, we rewrite the sentence: “Regards to energy metabolism, the hyperglycaemia and hyperinsulinemia observed in insulin resistant patients are also associated to elevated risk of cancer and a widely known mechanism of increasing insulin sensitivity and glucose uptake is upregulating AMPK activity (Miyamoto 2018).”

 Line 525, the authors indicate that AET-mediated lower M1, but higher M2 macrophages was observed in obese breast cancer patients, as compared with sedentary pairs. However, this is discordant with the above mentioned pathogenic  effects of M2 macrophages on cancer ( page 9, line 363). Therefore, these discordant results need to be discussed  

Author’s comments: Despite of this “paradox” in the context of M2 macrophages in cancer and inflammation, we want to clarify that the first citation (line 363 page 9) is related to tumour-associated macrophages polarization characteristics and subsequent pathophysiology. While the second citation (line 526 page 12) was focused on adipose tissue-associated macrophages from breast cancer survivors and its importance on the anti-inflammatory effect of exercise, compared to the sedentary group, especially on cancer recurrence. In other words, despite of the comparison between the same immune cell (macrophages) we are not comparing the same tissue (tumor versus adipose tissue).  With this we intend to highlight the important role of exercise on attenuating inflammation on adipose tissue, which is strongly related to breast cancer development and recurrence. However, this information that can lead to confusion, thus we highlighted these differences in the second citation. 

“Inflammation, systemically or locally, is another factor that is strongly modulated by exercise especially in the cancer context. Studies using a combination of aerobic and resistance exercise (16 weeks) in obese postmenopausal breast cancer survivors observed not only cardio metabolic and body composition progress but also anti inflammatory effects. Post exercise protocol, adipose tissue macrophages from the participants had lower M1 and higher M2 macrophages compared to their sedentary pairs. Additionally, IL-6 and TNF-α secretion also diminished, confirming the improvement on AT inflammation. Despite of the well known inflammatory tumour microenvironment in cancer patients and tendency to M2 macrophage polarization, these results suggest the importance of physical activity in chronic inflammation attenuation and on minimizing cancer recurrence (Dieli-Conwright, Parmentier et al. 2018).”

Ando, S., L. Gelsomino, et al. (2019). "Obesity, Leptin and Breast Cancer: Epidemiological Evidence and Proposed Mechanisms." Cancers (Basel) 11(1).

AsghariHanjani, N. and M. Vafa (2019). "The role of IGF-1 in obesity, cardiovascular disease, and cancer." Med J Islam Repub Iran 33: 56.

Bjorbak, C., H. J. Lavery, et al. (2000). "SOCS3 mediates feedback inhibition of the leptin receptor via Tyr985." J Biol Chem 275(51): 40649-40657.

Caslin, H. L., M. Bhanot, et al. (2020). "Adipose tissue macrophages: Unique polarization and bioenergetics in obesity." Immunol Rev 295(1): 101-113.

Chen, Y., Y. Song, et al. (2019). "Tumor-associated macrophages: an accomplice in solid tumor progression." J Biomed Sci 26(1): 78.

Dieli-Conwright, C. M., J. H. Parmentier, et al. (2018). "Adipose tissue inflammation in breast cancer survivors: effects of a 16-week combined aerobic and resistance exercise training intervention." Breast Cancer Res Treat 168(1): 147-157.

Grisouard, J., K. Dembinski, et al. (2011). "Targeting AMP-activated protein kinase in adipocytes to modulate obesity-related adipokine production associated with insulin resistance and breast cancer cell proliferation." Diabetol Metab Syndr 3: 16.

Jequier, E. (2002). "Leptin signaling, adiposity, and energy balance." Ann N Y Acad Sci 967: 379-388.

Jia, Q., M. E. Morgan-Bathke, et al. (2020). "Adipose tissue macrophage burden, systemic inflammation, and insulin resistance." Am J Physiol Endocrinol Metab 319(2): E254-E264.

Katira, A. and P. H. Tan (2015). "Adiponectin and its receptor signaling: an anti-cancer therapeutic target and its implications for anti-tumor immunity." Expert Opin Ther Targets 19(8): 1105-1125.

Lumeng, C. N., J. L. Bodzin, et al. (2007). "Obesity induces a phenotypic switch in adipose tissue macrophage polarization." J Clin Invest 117(1): 175-184.

Mauro, L., G. D. Naimo, et al. (2018). "Uncoupling effects of estrogen receptor alpha on LKB1/AMPK interaction upon adiponectin exposure in breast cancer." FASEB J 32(8): 4343-4355.

May, F. J., L. A. Baer, et al. (2017). "Lipidomic Adaptations in White and Brown Adipose Tissue in Response to Exercise Demonstrate Molecular Species-Specific Remodeling." Cell Rep 18(6): 1558-1572.

Miyamoto, L. (2018). "[AMPK as a Metabolic Intersection between Diet and Physical Exercise]." Yakugaku Zasshi 138(10): 1291-1296.

Naimo, G. D., L. Gelsomino, et al. (2020). "Interfering Role of ERalpha on Adiponectin Action in Breast Cancer." Front Endocrinol (Lausanne) 11: 66.

Nishimura, S., I. Manabe, et al. (2009). "CD8+ effector T cells contribute to macrophage recruitment and adipose tissue inflammation in obesity." Nature Medicine.

Nuri, R., M. Moghaddasi, et al. (2016). "Effect of aerobic exercise on leptin and ghrelin in patients with colorectal cancer." J Cancer Res Ther 12(1): 169-174.

Oh-ishi, S., T. Kizaki, et al. (1996). "Swimming training improves brown-adipose-tissue activity in young and old mice." Mech Ageing Dev 89(2): 67-78.

Shrestha, A., S. Nepal, et al. (2016). "Critical Role of AMPK/FoxO3A Axis in Globular Adiponectin-Induced Cell Cycle Arrest and Apoptosis in Cancer Cells." J Cell Physiol 231(2): 357-369.

Silveira, L. S., L. A. Biondo, et al. (2020). "Macrophage immunophenotype but not anti-inflammatory profile is modulated by peroxisome proliferator-activated receptor gamma (PPARgamma) in exercised obese mice." Exerc Immunol Rev 26: 10-22.

Simpson, K. A. and M. A. Singh (2008). "Effects of exercise on adiponectin: a systematic review." Obesity (Silver Spring) 16(2): 241-256.

Skurk, T., C. Alberti-Huber, et al. (2007). "Relationship between Adipocyte Size and Adipokine Expression and Secretion." The Journal of Clinical Endocrinology & Metabolism 92(3): 1023-1033.

Surmacz, E. (2000). "Function of the IGF-I receptor in breast cancer." J Mammary Gland Biol Neoplasia 5(1): 95-105.

Swisher, A. K., J. Abraham, et al. (2015). "Exercise and dietary advice intervention for survivors of triple-negative breast cancer: effects on body fat, physical function, quality of life, and adipokine profile." Support Care Cancer 23(10): 2995-3003.

Talukdar, S., D. Y. Oh, et al. (2012). "Neutrophils mediate insulin resistance in mice fed a high-fat diet through secreted elastase." Nat Med 18(9): 1407-1412.

Vosselman, M. J., J. Hoeks, et al. (2015). "Low brown adipose tissue activity in endurance-trained compared with lean sedentary men." Int J Obes (Lond) 39(12): 1696-1702.

Winer, S., Y. Chan, et al. (2009). "Normalization of obesity-associated insulin resistance through immunotherapy." Nat Med 15(8): 921-929.

Wu, M. V., G. Bikopoulos, et al. (2014). "Thermogenic Capacity Is Antagonistically Regulated in Classical Brown and White Subcutaneous Fat Depots by High Fat Diet and Endurance Training in Rats: IMPACT ON WHOLE-BODY ENERGY EXPENDITURE." Journal of Biological Chemistry 289(49): 34129-34140.

Yoshioka, K., T. Yoshida, et al. (1989). "Effects of exercise training on brown adipose tissue thermogenesis in ovariectomized obese rats." Endocrinol Jpn 36(3): 403-408.

Yu, N., Y. Ruan, et al. (2017). "Systematic Review and Meta-Analysis of Randomized, Controlled Trials on the Effect of Exercise on Serum Leptin and Adiponectin in Overweight and Obese Individuals." Horm Metab Res 49(3): 164-173.

Zulato, E., F. Bergamo, et al. (2014). "Prognostic significance of AMPK activation in advanced stage colorectal cancer treated with chemotherapy plus bevacizumab." Br J Cancer 111(1): 25-32.

Round 2

Reviewer 1 Report

I endorse the publication of this manuscript in its current form.

Reviewer 2 Report

Numerous  spelling mistakes are present in the manuscript